# Evolutionary Study of Disorder in Protein Sequences

**DOI:** 10.3390/biom10101413

**Published:** 2020-10-06

**Authors:** Kristina Kastano, Gábor Erdős, Pablo Mier, Gregorio Alanis-Lobato, Vasilis J. Promponas, Zsuzsanna Dosztányi, Miguel A. Andrade-Navarro

**Affiliations:** 1Faculty of Biology, Johannes Gutenberg University, Biozentrum I, Hans-Dieter-Hüsch-Weg 15, 55128 Mainz, Germany; kkastano@uni-mainz.de (K.K.); munoz@uni-mainz.de (P.M.); 2MTA-ELTE Momentum Bioinformatics Research Group, Department of Biochemistry, ELTE Eötvös Loránd University, H-1117 Budapest, Hungary; e.gabor90@gmail.com (G.E.); dosztanyi@caesar.elte.hu (Z.D.); 3Human Embryo and Stem Cell Laboratory, The Francis Crick Institute, 1 Midland Road, London NW1 1AT, UK; gregorio.alanis@crick.ac.uk; 4Bioinformatics Research Laboratory, Department of Biological Sciences, University of Cyprus, 2109 Nicosia, Cyprus; vprobon@ucy.ac.cy

**Keywords:** intrinsically disordered proteins, intrinsically disordered regions, comparative genomics, ortholog comparison

## Abstract

Intrinsically disordered proteins (IDPs) contain regions lacking intrinsic globular structure (intrinsically disordered regions, IDRs). IDPs are present across the tree of life, with great variability of IDR type and frequency even between closely related taxa. To investigate the function of IDRs, we evaluated and compared the distribution of disorder content in 10,695 reference proteomes, confirming its high variability and finding certain correlation along the Euteleostomi (bony vertebrates) lineage to number of cell types. We used the comparison of orthologs to study the function of disorder related to increase in cell types, observing that multiple interacting subunits of protein complexes might gain IDRs in evolution, thus stressing the function of IDRs in modulating protein-protein interactions, particularly in the cell nucleus. Interestingly, the conservation of local compositional biases of IDPs follows residue-type specific patterns, with E- and K-rich regions being evolutionarily stable and Q- and A-rich regions being more dynamic. We provide a framework for targeted evolutionary studies of the emergence of IDRs. We believe that, given the large variability of IDR distributions in different species, studies using this evolutionary perspective are required.

## 1. Introduction

Intrinsically disordered proteins (IDPs) contain intrinsically disordered regions (IDRs) that lack globular structure under physiological conditions, defying the traditional structure-function paradigm [1,2]. Often, these regions can be recognized by their peculiar amino acid composition [3], which has a tendency to present low complexity but that is not always the case [4]. IDPs evolve rapidly following mutational patterns different from those of globular proteins [5,6,7], participate in the formation of membraneless organelles in cells by liquid-liquid phase separation [8,9] and play key roles in cell-signaling, regulation and cancer [10]. Additionally, mutations within IDRs that increase the aggregation propensity of proteins have been directly linked to diseases such as Alzheimer’s, Parkinson’s, and Huntington’s disease [11]. In general, eukaryotic species are known to have more disorder than prokaryotes [3,12]. This, in addition to the fact that disorder provides an advantage for complex sets of interactions between signaling and regulatory proteins (see e.g., [13,14,15]) hints at the function of IDRs in facilitating the complex molecular networks required in eukaryotic and multicellular organisms [16,17]. However, the existence of single-celled species with high disorder content shows that there is still much to be learned in regards to intrinsic disorder function and evolution.

Here we present a two-step approach to better understand disorder from an evolutionary perspective that combines previous approaches (genome-wide analysis [3,12,18,19,20] and analysis of orthologs [21,22]), taking advantage of the increase in complete proteomes. First, we compare measures of disorder in 10,695 UniProt reference proteomes (complete proteomes). Afterwards we illustrate how a focused analysis using orthologs of human proteins present in all of four taxonomically distant species provides insights into the functions, composition and dynamic evolution of IDRs. Our approach can be flexibly applied to consider alternative measures of disorder and taxonomic focus.

## 2. Materials and Methods

For our proteome analyses, we used UniProt reference proteomes (completely sequenced proteomes selected manually and algorithmically from UniProt) to provide broad coverage of the tree of life. We used the UniProt release 2018_09 and selected only proteomes with at least 300 proteins (10,695). This allowed us to filter out species with very short proteomes that could statistically bias our comparison. Viruses were the species with the fewer proteins; only 81 out of around 6000 Viruses had more than 300 proteins and thus were kept for the analysis. This cut-off includes most of the known giant virus families [23].

The predictions of IDRs were done with IUPred2A [24]. IUPred2A uses inter-residue statistical interaction potentials and for each input protein sequence outputs a score between 0 and 1 for each residue. This score represents the probability that the residue is part of a disordered region and we used the default 0.5 as the cut-off. We considered regions of at least 30 consecutive disordered residues, that is, all disordered regions predicted had a minimum length of 30 residues. Compositionally biased regions (CBRs) were predicted using CAST [25] with default settings (BLOSUM62 mutation matrix, threshold 40). A protein was considered to be an IDP if it contained at least one IDR.

For the comparison of disorder across orthologs we obtained the 984 pairs of one-to-one orthologs from the InParanoid database (latest version: 8.0, released in December 2013 [26]) with proteins in all five species of interest: *Mus musculus*, *Danio rerio*, *Drosophila melanogaster*, and *Saccharomyces cerevisiae*. The subset of 281 groups where the human protein was an IDP was used for analysis (see Appendix A) and the complete dataset was used as background.

To evaluate the conservation of disorder within groups of orthologous sequences we used the following protocol. Given a group of orthologous sequences, we first aligned the protein sequences using MUSCLE [27]. Then, we used the resulting multiple sequence alignment to align the presence of disorder in each pair of species. This allowed us to find the positions in which disorder was present in both proteins of an orthologous pair and the positions where it was present in only one of them. Next, we evaluated the conservation of disorder using the Jaccard similarity index (number of disordered residues in the same position in the two species divided by the number of disordered residues anywhere in any of the two species). To be sure that the sequence identity between the orthologs was not responsible for the disorder conservation, we verified that there was no correlation between disorder conservation and sequence identity values (*r* = 0.33 for pairs of orthologs between human and mouse, 0.27 for human-fish, 0.19 for human-fly and 0.12 for human-yeast).

To find common patterns of disorder conservation, we performed hierarchical clustering (Euclidean distance matrix and complete linkage; R functions dist and hclust) of the orthologous groups according to the four conservation scores between human and mouse, fish, fly and yeast, respectively. To obtain clusters, we cut the dendrogram using a cut-off point that avoided having many clusters with very few proteins and resulted in homogeneous clusters when visually analyzing the resulting heat map. The heat map was produced with the R package Complex heatmap [28] and the clusters were created after cutting the dendrogram with the function cuttree.

Functional enrichment analysis of the human genes in these clusters was done with PANTHER 15.0 with GO-slim annotations and Fisher’s exact test with the Bonferroni correction for multiple testing [29]. Enrichment was calculated using the complete set of 984 genes with orthologs in the five species of interest as background.

## 3. Results

### 3.1. Distribution of Disorder in Complete Proteomes

To have an overview of the variation of disorder content in the proteins of different species (10,695 completely sequenced proteomes; only viruses with large proteomes included; see Section 2 for details) we classified the data in taxonomic divisions.

We considered Viruses (81 species), Archaea (464), Bacteria (8968), and a series of groups within Eukaryota: Euarchontoglires (35), Euteleostomi (excluding Euarchontoglires, 103), Metazoa (excluding Euteleostomi, 199), Fungi (602), and (the rest of) Eukaryota (199). Of note, the over-representation of bacterial proteomes is due to the high number of bacterial sequencing projects, which are facilitated by the accessibility and small genome size of bacteria [30].

To quantify disorder, we used the fraction of disordered proteins for each species (IDP fraction; Figure 1a; see Material and Methods for definition of disordered protein) and the fraction of disordered residues in the whole proteome (IDR fraction; Figure 1b) (IDRs predicted using IUPRed2A [24]; see Section 2 for details). IDP and IDR fractions follow similar trends. As it is already well known [3,12], Archaea and Bacteria have lower disorder content than Eukaryota on average. Within Eukaryota, disorder content generally increases in divisions that include species with more cell types. However, Fungi and the rest of Eukaryota have median values higher or comparable to Metazoa and Euteleostomi (bony vertebrates).

For a more detailed analysis of the latter observation, we contrasted unicellular versus multicellular eukaryotes, with bacteria shown for comparison (Figure 1c). In this case, we compared Viridiplantae (105 species), Fungi (602), Alveolata (62), Euglenozoa (19), Metazoa (337) and the rest of Eukaryota (48). Bacteria (8968) are shown for comparison (Figure 1c). The distribution of values (red dots) indicate great variability within each group, and, most importantly, that there are a significant number of unicellular organisms that have more disorder in their proteomes than any of the Metazoan species considered. In fact, it is interesting to note that there are bacterial species that reach average Metazoan values.

These results suggest that while there might be a relation between disorder content and organism complexity in particular taxonomic realms (e.g., Metazoa < Euteleostomi < Euarchontoglires or Prokarya < Eukaryota; Figure 1a,b), this relation cannot be generalized and great variability of disorder content is observed for given taxonomic divisions.

Another observation is that IDP and IDR fractions seem to provide rather similar results (Figure 1a,b). We explore in the next sections how these two measures correlate and whether using one or the other leads to different conclusions about the use of disorder in protein sequences.

### 3.2. Distribution of Disorder throughout Five Main Taxa

We first compared the two measurements of disorder we are using. A strong correlation was found between the IDP and the IDR fractions in species from different taxa (Figure 2a) (*r* = 0.97 for the complete set).

Since disorder in protein sequences has been associated to the emergence of cellular complexity [11] we wondered if there could be a correlation between fraction of disorder and proteome size, which is a possible measure of organismic complexity, following [12]. Neither IDP nor IDR fractions correlate to number of proteins in an organism. In fact, within Bacteria, Fungi or Eukaryota, the species with the highest content of disorder have average proteome sizes for the corresponding taxa (Figure 2b; Appendix A).

In Viruses, *Pandoravirus dulcis* stands out (IDP fraction = 0.48; IDR fraction = 0.14) with values above any Bacteria or Archaea, even higher than those of most Eukaryotic species. In Fungi and rest of Eukaryota, 14 and 6 species, respectively, stand out from the central cloud of data points with IDP fraction > 0.6 and IDR fraction > 0.2. Of these, six are Basidiomycota in Fungi, and, intriguingly, in the rest of Eukaryota we find 14 species all of which produce spores: 12 species of Apicomplexa, the choanoflagellate *Salpingoeca rosetta* (strain ATCC 50,818/BSB-021) and the single-cell green alga *Chlamydomonas reinhardtii* [31]). In Archaea there is a visibly distinct group containing species with high IDP and IDR fractions compared to the rest (IDP fraction > 0.2, IDR fraction > 0.05). Most species in this group belong to the Halobacteria class (Appendix A). This result has been previously explained as a result of halobacteria adaptation to a high salinity environment [32,33].

### 3.3. Correlation of Disorder with the Number of Cell Types

Alternative to proteome size, the number of different cell types or tissues in an organism has also been used as a measure of organismic complexity (see e.g., [12,20]). Thus, we studied the correlation between IDP or IDR fraction and number of cell types for the set of 44 species that were annotated for this property in [12] (Figure 3). Species having less than 100 different cell types (Fungi, Plants, some Metazoa) are quite heterogeneous and do not present any correlation between disorder and number of cell types. However, when it comes to all Euteleostomi (including Euarchontoglires) there is a tendency for disorder to increase with number of cell types (IDP fraction correlation *r* = 0.56, *p*-value = 0.04, IDR fraction correlation *r* = 0.71, *p*-value = 0.006). If we also include Metazoa, the correlation between IDP fraction and number of cell types becomes stronger (*r* = 0.80), but the correlation between IDR fraction and number of cell types decreases (*r* = 0.50). Using the whole dataset further decreases correlation values (*r*-values of 0.17 and 0.19 for IDP and IDR, respectively).

### 3.4. Emergence of Intrinsic Disorder and Protein Function

Our general study above discussed a possible relation between disorder and organismic complexity. However, examples such as *Halobacteria* suggest that there are likely other functions in disordered regions that could be related to environmental factors. We also confirmed that there is an increase of disorder content in Euteleostomi species following their increase in cell type number (Figure 3). We hypothesized that a study focused on the evolutionary emergence of disorder associated with an increase in cell type number using selected species could reveal the functions of IDRs associated with organismic complexity. For this, we propose a protocol that allows such an evolutionary study of the function of disorder by comparing the IDPs of a reference species against sets of orthologs from selected species.

Here, we use human as reference species and compare its IDPs to the corresponding orthologs in four well-studied species with various tissue numbers: *M. musculus*, *D. rerio*, *D. melanogaster*, and *S. cerevisiae*. We obtained a total of 281 orthologous groups of a human IDP with proteins in all other four species (see Appendix A; see Section 2 for details).

We evaluated the conservation of disorder between each human protein and the corresponding orthologs using the multiple sequence alignments of each orthologous group. For this we followed a protocol that provided a score of disorder conservation between the human sequence and each of the orthologs in the other four species with values from zero (indicating that the predicted disordered regions of the sequences did not align to each other) to one (indicating that the predicted disordered regions fully aligned) (see Methods and Materials for details). We performed hierarchical clustering of the conservation scores between human and the four other species across all orthologous groups (Figure 4; see Section 2 for details). From these data, we defined seven clusters of orthologous groups with the following properties (from top to bottom in Figure 4): (1) a cluster with a single orthologous group, (2) a cluster with orthologous groups of sequences with high conservation to human for all species, (3) high conservation to human for the four species except *Drosophila*, (4) high conservation to human for the four species except for baker’s yeast, (5) low conservation to human for all species, (6) high conservation to human only for mouse, and (7) high conservation to human for mouse and *Drosophila*. These patterns of similarity correspond well to the fraction of proteins in the cluster for each species that has disorder (Figure 5). For example, cluster #3 has low conservation of disorder between human and *Drosophila* and accordingly the IDP fraction for the *Drosophila* proteins in this cluster is slightly below 0.50, while the other species have IDP fractions of one or close to one.

IDPs are known to be generally enriched in signaling and regulatory functions with many of them functioning as kinases, splicing factors and transcription factors [15,16]. To investigate if these functions distribute differently across the common patterns of disorder conservation observed above, we did a functional enrichment analysis of the human genes in each cluster (see Section 2 for details). In general, we observed functions situated in the nucleus and very few terms associated to particular physiological processes: most functions have to do with the organization and maintenance of the genetic material, the biogenesis of ribosomal rRNAs, and transcription, but rarely splicing or cell cycle control (Figure 4) (Appendix A
Appendix A). While all clusters had functions related to the nucleus, there are particular nuclear functions differently enriched that we briefly summarize in the next paragraphs.

Cluster #2 contains proteins with disorder conserved in all species, thus established long ago and preserved since then. This is the only cluster enriched in functions related to the nucleolus. Enriched functions relate to interactions with DNA and poly-adenylated RNA, particularly DNA transcription by the subunits of several RNA polymerases. For example, we have DNA binding subunits RPABC2, POLR1A, POLR3D and ERCC3 (UniProt AC P61218, O95602, P05423, P19447, respectively). There are also a number of proteins that activate RNA polymerase II such as GTF2E2, SUPT6H and RTF1 (P29084, Q7KZ85, Q92541). Other DNA binding functions observed here include three subunits of the MCM complex, the helicase complex used once per cell cycle, MCM2, MCM3 and MCM4 (P33991, P49736, P25205).

Cluster #4 is the largest cluster and consists of proteins with disorder conserved in the species closest to human, with lower conservation in *Drosophila* and yeast. This cluster is enriched in proteins associated to chromosomes involved in chromatin remodeling: SMARCAD1, INO80, DMAP1 and VPS72 (Q9H4L7, Q9ULG1, Q9NPF5, Q15906). We collected experimentally-known protein interactions between these proteins using the HIPPIE database [34] and found that DMAP1 interacts both with VPS72 (as part of the SRCAP complex, which mediates the exchange of histone H2AZ1/H2B dimers for nucleosomal H2A/H2B) and with SMARCAD1. This cluster is also enriched in ribonucleoproteins, particularly of components of the tUTP complex, a subcomplex of the small subunit (SSU) processome, which mediates the processing of the 18S rRNA subunit.

Cluster #6 contains sequences that are conserved in mammals, human and mouse, like cluster #6 above, but less conserved in zebrafish. They show enrichment in chromosomal associated proteins, including an enrichment in members of DNA helicase complexes: pleiotropic regulator 1 (PLRG1), a component of the spliceosome, subunit of the INO8 chromatin remodeling complex ACTR8, and Replication protein A 70 KDa DNA-binding subunit (RPA1), which is a subunit of the heterotrimeric replication protein A complex (O43660, Q9H981, P27694).

Cluster #5 represents human proteins with different disorder content than their orthologs in all the other species considered. This cluster contains proteins that are related to chromosome condensation: DNA repair protein RAD50, and two subunits of the condensin complex, NCAPG, NCAPH (Q92878, Q9BPX3, Q15003). We find as well three proteins annotated with the GO term “translesion synthesis”: REV3L, REV1 and POLH (DNA polymerase eta, Polη) (O60673, Q9Y253 and Q9UBZ9). According to the HIPPIE database, these proteins are known to interact with each other [35,36]. These results suggest that disorder evolution might be tied between proteins that interact, which is in agreement with a general function of disorder in the modulation of protein interactions.

All the clusters considered above correspond to patterns of disorder conservation coherent with a single evolutionary event of inclusion of disorder in the family occurring at some point in the lineage towards the human protein. We discussed these clusters starting from those considering a further event (occurring before the divergence of fungi from the human lineage, conserved in all the species considered), to a closer event (occurring after the divergence of mouse from the human lineage, not conserved in any of the species considered). Other patterns of conservation are possible but only those frequent and strong enough are reflected in the clustering. In our analysis, one such pattern is reflected by cluster #7, which displays conservation of disorder to human that is higher in mouse and *Drosophila*, and lower in *Danio* and yeast. Ribosome biogenesis related functions dominate this cluster. Using the HIPPIE database, we found that three proteins in this cluster related to ribosome biogenesis are known to interact: we identified SHQ1 (Q6PI26), which interacts with NAF1 (Q96HR8) to generate H/ACA RNPs, and GAR1 (Q9NY12), a subunit of the H/ACA snoRNP complex, which also interacts with SHQ1. Our results seem to suggest that a significant number of proteins related to the H/ACA complex are more similar to human in *Drosophila* than in *Danio* in regard to their content of disordered regions. Finding out if this would lead to functional and mechanistic differences would require further investigation.

The last cluster we discuss is cluster #3, the smallest of the clusters but strong enough for detection. In this cluster, disorder is conserved for all species, except for *Drosophila*. Thus, this cluster is different from the other clusters observed because it contains protein families where evolutionary pressure specific to the lineage leading to *Drosophila* made them adopt a different disorder distribution in their sequences. Like the previous cluster, this small cluster is dominated by functions related to ribosome biogenesis, including TSR2 and probable ATP-dependent RNA Helicase DHX37, both annotated as involved in the maturation of the SSU-rRNA (Q969E8, Q8IY37).

The functional enrichment analysis shown above was performed on the human genes. If the functional enrichment is done for the *Drosophila* of for the yeast orthologs, we observe very few differences suggesting that the functionality of these families is well conserved over large evolutionary distances (Appendix A and Appendix A).

A few examples described above suggested the presence of interacting subunits of protein complexes in some clusters. To test if this result is significant, we compared the number of observed protein-protein interactions (PPIs as recorded in the HIPPIE database [34]) between the human proteins in each cluster to the number of PPIs obtained between similarly sized sets of proteins randomly taken from the background of 984 human genes with orthologs. The result indicates a significant result for clusters #2 and #7 (*p*-value < 0.05; Table 1). We repeated the analysis for mouse PPIs (using the MIPPIE database [37]) but due to the lower numbers of reported interactions for this species, no significant results were observed (data not shown).

### 3.5. Compositionally Biased Regions in Orthologs

Disordered regions have different properties according to composition, which might correspond to different properties and functions [16]. For example, their distributions of charged residues can affect their level of compaction [38], conformational preferences [39] and phosphorylation [40], which ultimately influence the propensity of IDPs to aggregate [41,42,43], interact with other proteins [44] and perform phase separation [45]. To complement our analysis of disorder emergence in orthologs presented above, we evaluated the presence of compositionally biased regions (CBRs) in these proteins (using CAST [25]). CAST reports the existence of regions where the frequency of one amino acid is unusually high. These regions can overlap.

Disordered regions can occur without composition bias but often CBRs and disordered regions overlap [4]. Accordingly, we observe that some species differ in the number of proteins identified to have CBRs and IDRs. In particular, while the proteomes of the rest of the Eukaryota and fungi have IDP fractions comparable to those of Metazoa, they have lower fractions of proteins with CBRs than Metazoa (Figure 6a). This suggests that disorder not based on local composition bias is comparatively more frequent in non-metazoan eukaryotes. Within the archaea group, the Halobacteria group is not so clearly segregated using CBRs compared to IDPs or IDRs (Appendix A) suggesting that the compositional bias of the disorder typical of Halobacteria is moderate. In any case, there is a good correlation between fraction of proteins with CBRs and IDP fraction within each large taxonomic group (Figure 6b).

We next studied the patterns of CBR presence by amino acid category in the set of orthologs conserved in five species (984 groups of orthologs), which was used above to select orthologs for the study of the conservation of disordered regions in human proteins (281 groups of orthologs). We use binary patterns for representation. For example, given one of the 984 groups of orthologs, if Q-rich CBRs are detected in the orthologs of all the five species (from *H. sapiens*, *M. musculus*, *D. rerio*, *D. melanogaster* to *S. cerevisiae*), irrespective of CBR position and number of regions detected, then the pattern of conservation of Q-rich regions in this group of orthologs is ‘11111′. If Q-rich regions are detected only in *H. sapiens* and *M. musculus* but not in any of the other three species orthologs, then the pattern is ‘11000′. And so on.

The frequency of patterns is indicative of how CBRs are generally conserved across the five species analyzed (Figure 7a). Again, these results depend strongly on the group of species that we have chosen. Nevertheless, we can appreciate that patterns with absence in one species surrounded by presence (patterns containing ‘101′) are very rare; these patterns would account mostly for either separate emergence of X-rich regions (where ‘X’ refers to the overrepresented amino acid type) in the taxonomic divisions leading to the corresponding species, or, alternatively, for the emergence of the X-rich region in an ancestral species followed by its loss in another species. Three of the four most common patterns are the presence of X-rich regions in only one of the five species (*D. melanogaster*, *S. cerevisiae* and *D. rerio*, respectively); the presence of the X-rich regions in the five species is the third most frequent pattern (Figure 7a).

If we study pattern conservation by the type of X-rich region, we can see strong differences between the most common CBRs. For example, for E-rich and K-rich CBRs among the most frequent patterns, we see those implying emergence once and conservation of the CBR (‘11111′, ‘11110′, ‘11100′). In Q-rich and A-rich CBRs we do not see these patterns frequently and instead, we see patterns implying that the evolution of these is more dynamic and occurs often in single species (‘00001′, ‘00010′, ‘00100′) (Figure 7B; Appendix A).

Figure 8 shows the frequency of the presence-absence profiles for all the CBR types as a heatmap. As described above, we can see that there are E-, K- and S-rich CBRs that are often present in all 5 species of the ortholog groups (11111). Looking for species-specific patterns we can see that N-rich CBRs stand out as specific to yeast: 00001 is the most frequent pattern for this type of CBR. This is consistent with [46] where it was found that homorepeats (CBRs consisting of repetitions of a single amino acid residue) of type N and lengths 6-20 are more common in yeast than in human. For *D. melanogaster* specific regions (00010), the most frequent type is Q-rich, and this pattern happens to be also the most frequent for Q-rich CBRs.

These results highlight the evolutionary differences between different types of X-rich CBRs, suggesting that, in the set of species compared, E- and K-rich CBRs are stable, while Q- and A-rich CBRs are more dynamic. This is probably related to the functions of CBRs, and by extension of IDRs, related to composition. Once more, we remark that the results presented in this section are highly dependent on the choice of species. CBR usage can be very particular for a given species (and by extension for a taxon), as exemplified by the specific enrichment in N-rich CBRs for *S. cerevisiae*.

## 4. Discussion

In this work we have approached the study of disorder in protein sequences following an evolutionary perspective [2,5,6,17] including strategies that compare disorder usage across groups of orthologs [7,22]. Our study of disorder in proteomes allowed us to illustrate the general tendencies of disorder distribution. We observed the change of disorder content throughout various taxonomic groupings. We found a tendency of disorder to increase from Metazoa along Euteleostomi to Euarchontoglires and a correlation between disorder content and number of cell types in all Euteleostomi. However, due to several fungi, alveolates and plants having values of relative disorder higher or comparable to Metazoa and Euteleostomi, we found that this is not a general relation.

To study the association of protein function to the evolutionary emergence of disorder in relation to increased number of cell types, we used a focused approach comparing orthologs of human proteins with disorder to their corresponding orthologs in four species with various numbers of cell types (mice, fish, fly and yeast). We aligned the groups of orthologous sequences and consequently the disordered regions, and produced disorder conservation scores between each species and human. This allowed us to cluster genes with particular IDR conservation patterns in the different species and associate functions to them with a functional enrichment. We found that the most conserved functions in similarly disordered proteins across the five species studied are related to the nucleus. The nucleus uses large machineries for the management of the genetic material and for control of transcription. It is possible that because IDRs are often involved in the modulation of protein interactions, we find signals from the co-evolution of interacting disordered regions emerging together in interacting partners. The number of interactions among human proteins in the largest cluster was significantly above expectation (Table 1). Similarly, co-evolution of coiled coil regions was previously found in the orthologs of interacting human proteins [47].

Because compositionally biased regions (CBRs) are well correlated to IDRs and are known to influence their structural and functional properties, for example with particular distributions of charged residues [39,40,41,42,44,45], we also used them in the ortholog analysis to characterize amino acid biases in IDPs. Biased regions behaved very differently with Q- and A-rich regions being highly species-specific and E- and K-rich ones being very conserved. We provided evidence that local compositional biases tend to evolutionarily emerge and disappear in IDPs in residue type-specific manners. However, further analyses are necessary to identify correlations of local compositional features to the functional properties of IDPs in general and of IDRs in particular.

## 5. Conclusions

Taken together, our approaches show different strategies that can be used to study disordered regions using comparative genomics. On the one hand, the abundance of complete proteomes allows exploring and comparing the distribution of different measurements of disorder regions. On the other hand, we have shown how it is possible to use large sets of orthologs from a few selected species to focus the analysis. The genome-wide analysis is important to tune and quantify the measures of disorder to be employed, while the ortholog analysis brings species-specific functional insights. In our example, we used five eukaryotic species taxonomically very distant (from fungi to human). Many proteins that do not have orthologs in all these species (e.g., brain-related) are necessarily missing from the analysis. If a different focus is required, using the same approach, one can change the focus by analyzing sets of species more taxonomically related (e.g., primates). This avenue is likely to give insights into the context and functionality of IDRs in many different species.

## Figures and Tables

**Figure 1 biomolecules-10-01413-f001:**
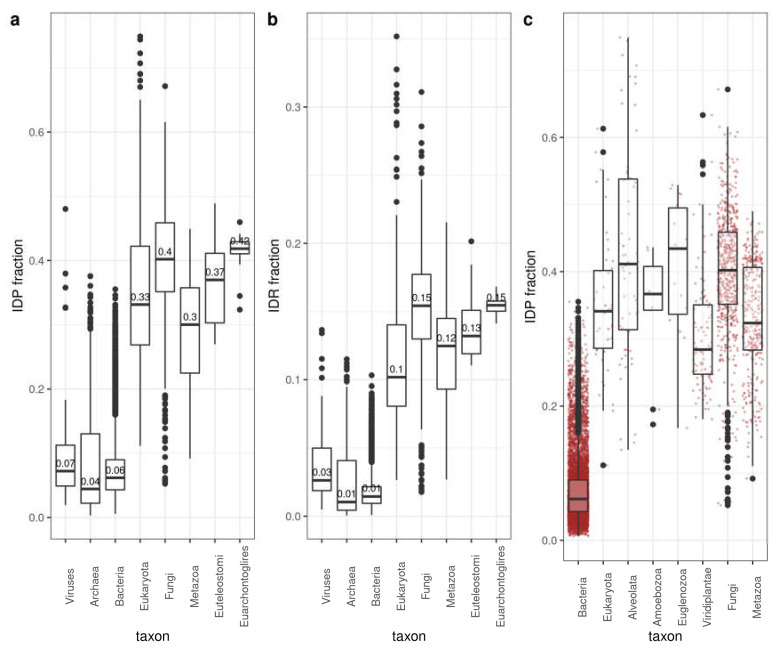
Disorder content by taxa. (**a**) IDP fraction. Number of proteins with at least one IDR divided by the number of proteins in the corresponding proteome. (**b**) IDR fraction. Number of residues in the corresponding proteome divided by the total number of residues in the corresponding proteome. (**c**) IDP fraction variability within eukaryotic groups (bacteria shown for comparison). Disorder values as computed by IUPred2A. See Section 2 for details. Corresponding distributions of values are represented as box-plots. Boxes are drawn from the first to the third quartile with whisker length 1.5 times the interquartile range.

**Figure 2 biomolecules-10-01413-f002:**
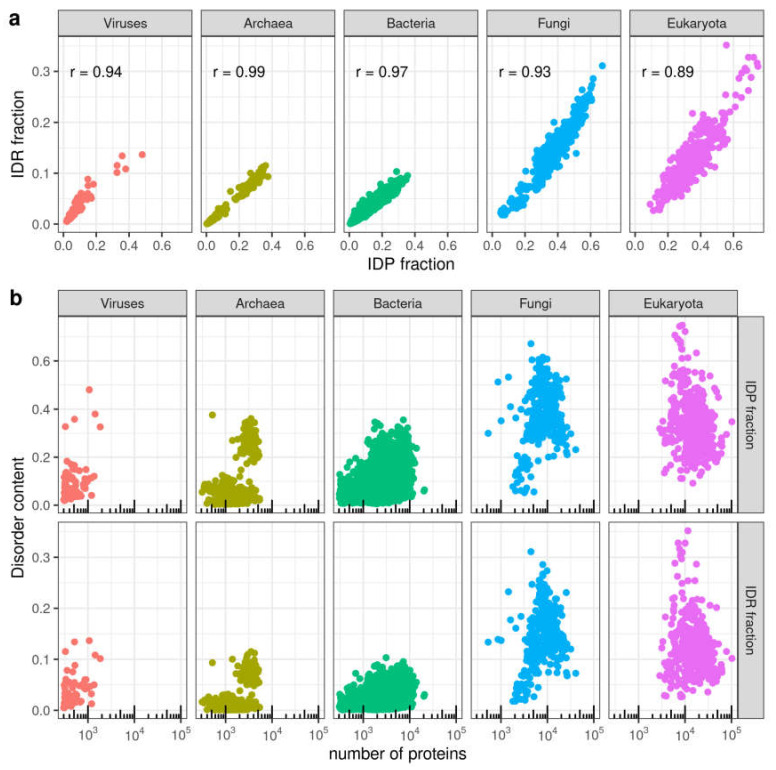
Correlations between measurements of disorder and protein content in genomes. (**a**) Relationship between IDR fraction and IDP fraction. (**b**) Relationship between disorder content (IDP fraction or IDR fraction) and number of proteins. Disorder values as computed by IUPred2A. See Section 2 for details.

**Figure 3 biomolecules-10-01413-f003:**
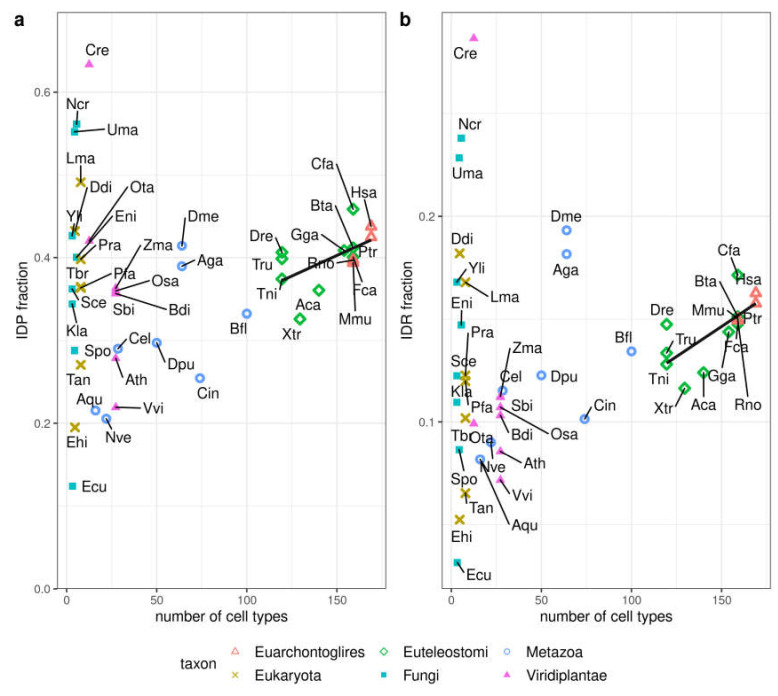
Disorder content vs. number of cell types. Disorder in 44 species measured as (**a**) IDP fraction and (**b**) IDR fraction. The number of cell types was obtained from [12]. The line indicates the linear fit for Euteleostomi (bony vertebrates) species (see text for details). See Appendix A for the abbreviations used for the species names.

**Figure 4 biomolecules-10-01413-f004:**
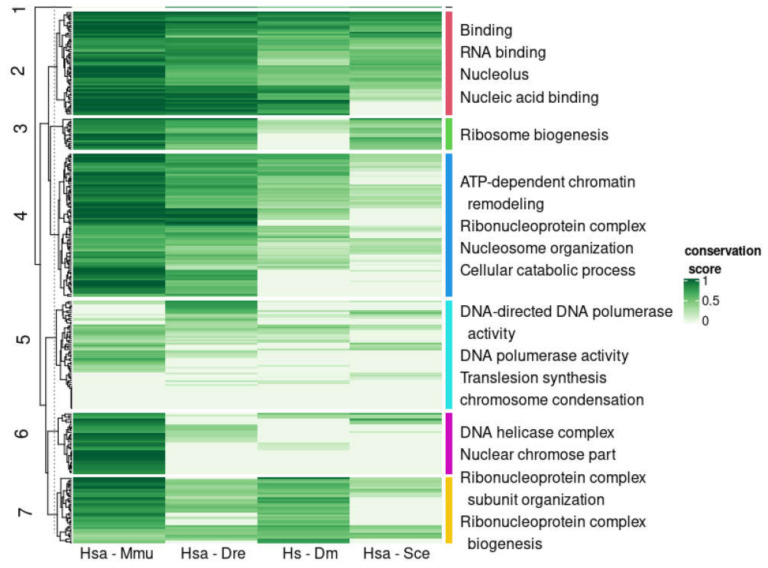
Similarity in disordered regions between human sequences and orthologs in four species. Columns indicate the similarity from high (dark green) to low (white) between the ortholog in the human protein and the mouse, fish, fly, and yeast ortholog, in disorder content (Jaccard index, see text for details). Enriched Gene Ontology terms of the corresponding human proteins in each cluster are indicated to the right (non-redundant manual selection; see full list and associated genes as Appendix A). See Section 2 for details.

**Figure 5 biomolecules-10-01413-f005:**
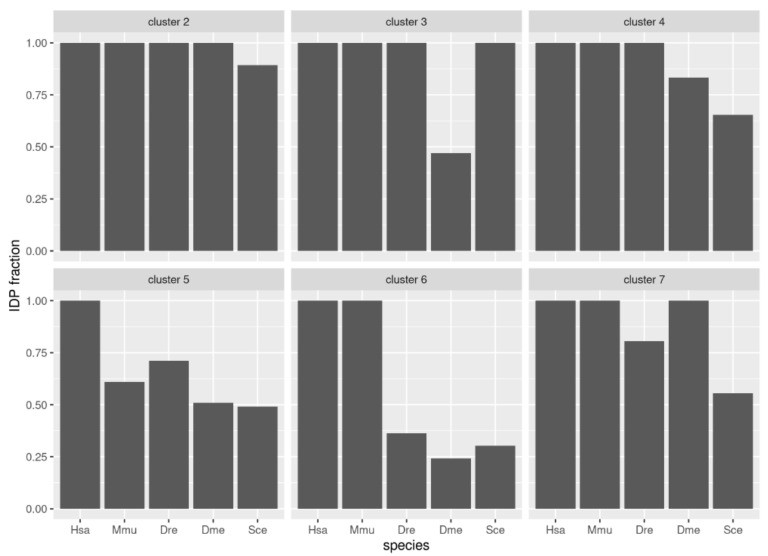
Disorder content in groups of proteins from five species. The groups correspond to the clusters indicated in Figure 4.

**Figure 6 biomolecules-10-01413-f006:**
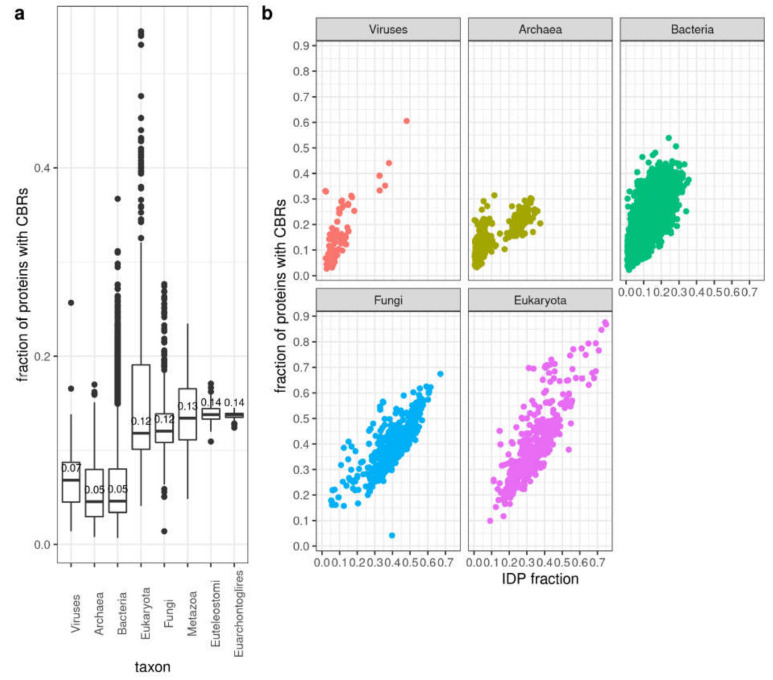
Protein CBR content. (**a**) By taxa. Box-plots are represented as in Figure 1. (**b**) Fraction of proteins with CBRs versus fraction of proteins with IDPs. Pearson correlation coefficient (*r*) values are: Viruses = 0.78, Archaea = 0.79, Bacteria = 0.78, Fungi = 0.75, Eukaryota = 0.82 (all with *p*-value < 2e-16). CBRs computed by CAST. See Section 2 for details.

**Figure 7 biomolecules-10-01413-f007:**
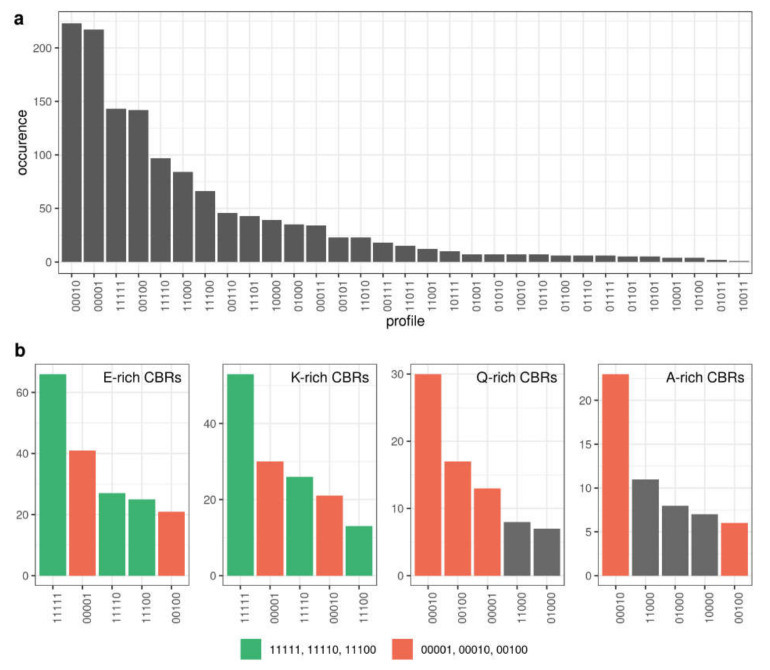
Number of CBRs per conservation pattern. (**a**) All CBRs (**b**) Top 5 most frequent patterns for E-rich, K-rich, Q-rich and A-rich CBRs. Patterns coloured in green indicate generation once and conservation since (‘11111′, ‘11110′, ‘11100′), and patterns coloured in red indicate specific presence in a species (‘00001′, ‘00010′, ‘00001′). Other patterns are coloured in grey. The five-digit binary conservation patterns refer to the presence (1) or absence (0) of the corresponding CBR-type in each of the five proteins from an orthologous group for the following species: *H. sapiens*, *M. musculus*, *D. rerio*, *D. melanogaster* and *S. cerevisiae* (see text for details).

**Figure 8 biomolecules-10-01413-f008:**
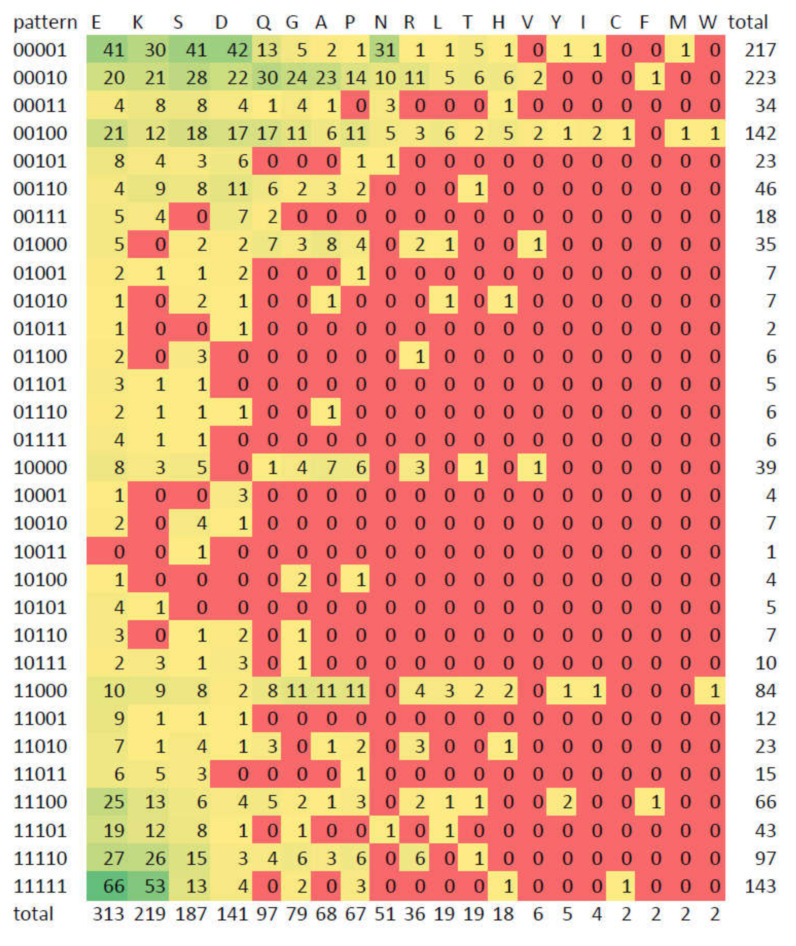
Numbers of CBRs by type and pattern. The heatmap indicates frequency from green (high) to red (low). Also available as Appendix A. See caption of Figure 7 for the definition of the five-digit binary conservation patterns.

**Table 1 biomolecules-10-01413-t001:** Number of PPIs between human proteins in clusters. Columns indicate: cluster number (c), number of protein-protein interactions observed among the proteins in the cluster (PPI), average simulated value (expect), standard deviation of simulated values (stdev), *z*-score of the observed value, and *p*-value of the observed value. See text for details.

c	PPI	Expect	Stdev	*z*-score	*p*-value
2	68	35.76	10.90	2.96	0.0015
3	3	3.21	2.27	−0.09	0.536
4	64	69.79	15.47	−0.37	0.646
5	25	39.87	11.27	−1.32	0.906
6	6	12.33	5.30	−1.19	0.884
7	27	14.84	6.05	2.01	0.022

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
