# Peer review of "Evolutionary Study of Disorder in Protein Sequences"

_biomolecules, 2020, doi:10.3390/biom10101413_

Round 1
Reviewer 1 Report
In the manuscript entitled: "Evolutionary study of disorder in protein sequences" Andrade-Navarro and colleagues apply a suite of relatively standard sequence analysis tools to investigate the evolution of protein disorder. Overall I think the analysis presented is fine but some of the figures are not clear and some interpretations should be revised.
Your definition of IDR fraction is not clear. On page 2 line 77 you say it is the fraction of disordered residues in the whole proteome but in the methods you say it is a disordered segment of at least 30 residues. Are both of these things included in the calculation? Is IDR fraction = The number of residues in disordered segments that are at least 30 consecutive residues divided by the total number of residues in the proteome? If this is the case then the difference between IDP fraction and IDR fraction may be related to the genomic economy of IDRs. They do as much as an ordered protein but require fewer amino acids.
For Figure 1 you need to clarify how IDP and IDR fraction are calculated. Is IDP fraction = the number of proteins with at least one IDR divided by all proteins? What is IDR fraction. Is IDR fraction equal to the number of IDRs divided by all proteins? Is IDR length considered in the calculation.
On page 3 line 101 the authors state: "Another observation is that IDP and IDR fractions provide different results (Figure 1a-b). This suggests that even when using a single method to predict disorder, the interpretation of the data might matter when evaluating disorder content."
This isn't a very compelling or satisfying analysis. Oh well IDP and IDR fractions are different so lets blame IUPred. Isn't this what you expected? See the previous comments. Since you brought it up you probably need to repeat the calculations using at least two other disorder predictors and compare the results.
If my ealier critique is accurate then the analysis presented in Figure 2 is superfluous and should be moved to the supplement. You also have to reference your definition of complexity and provide more context for why you chose to focus on genome size. When is it appropriate to use proteome size as a measure of complexity? What is your definition of complexity? When is the number of proteins in the proteome NOT a measure of increased information content or processing?
The correlation of disorder with the number of cell types also seems superficial and overlooks the main point of the data. IDP fraction and IDR fraction have the same dependence (slope) on the number of cell types. Is this an accident of the way the two fractions are calculated or does it have some biological meaning?
On page 6 line 148 the authors state: "The results from our general study above indicate that while there might be a relation between disorder and complexity, there are likely other functions in disordered regions related to environment or ecology."
I don't know about this. Seems like a leap. There is some relation but the variability in cellular environment can reinforce, cancel, or offset variability in ecology.
On page 6 lines 156-173 should be moved to the methods.
The meaning of page 6 lines 186-188 is not clear. What does this mean? Which species have IDP fractions close to 1? This is not apparent from the data in Figure 1. According to the definition of IDP fraction you used in Figure 1 a value of 1 would mean that every protein in the proteome has a >/= IDR of 30 residues.
Author Response
We thank the referee for the insightful comments, which have certainly helped to improve the manuscript.
In the manuscript entitled: "Evolutionary study of disorder in protein sequences" Andrade-Navarro and colleagues apply a suite of relatively standard sequence analysis tools to investigate the evolution of protein disorder. Overall I think the analysis presented is fine but some of the figures are not clear and some interpretations should be revised.
Your definition of IDR fraction is not clear. On page 2 line 77 you say it is the fraction of disordered residues in the whole proteome but in the methods you say it is a disordered segment of at least 30 residues. Are both of these things included in the calculation? Is IDR fraction = The number of residues in disordered segments that are at least 30 consecutive residues divided by the total number of residues in the proteome? If this is the case then the difference between IDP fraction and IDR fraction may be related to the genomic economy of IDRs. They do as much as an ordered protein but require fewer amino acids.
We are sorry for the confusion. The method used for disorder prediction has a cut-off so that the minimum length of a region predicted to be disordered is 30 residues. No predicted disordered region had a shorter length. We have clarified this in the now expanded methods section.
For Figure 1 you need to clarify how IDP and IDR fraction are calculated. Is IDP fraction = the number of proteins with at least one IDR divided by all proteins? What is IDR fraction. Is IDR fraction equal to the number of IDRs divided by all proteins? Is IDR length considered in the calculation.
We apologize again for the lack of clarity. The referee is right about the IDP fraction but the IDR fraction is the fraction of residues in IDRs in the corresponding proteome. We have added the explicit definitions of IDP fraction and IDR fraction to the caption of Figure 1.
On page 3 line 101 the authors state: "Another observation is that IDP and IDR fractions provide different results (Figure 1a-b). This suggests that even when using a single method to predict disorder, the interpretation of the data might matter when evaluating disorder content."
This isn't a very compelling or satisfying analysis. Oh well IDP and IDR fractions are different so lets blame IUPred. Isn't this what you expected? See the previous comments. Since you brought it up you probably need to repeat the calculations using at least two other disorder predictors and compare the results.
With this sentence we did not want to imply that there was a significant difference. In fact, we argue otherwise a few lines below: “A strong correlation was found between the IDP and the IDR fractions in species from different taxa”. We see now that the sentence mentioned by the reviewer gives the wrong impression and we changed it to avoid appearing to suggest that there is a problem with IUPred or that we prefer IDP to IDR fraction.
If my ealier critique is accurate then the analysis presented in Figure 2 is superfluous and should be moved to the supplement.
We hope that with the clarification about the meaning of IDP fraction and IDR fraction the reviewer will appreciate that Figure 2 is not superfluous.
You also have to reference your definition of complexity and provide more context for why you chose to focus on genome size. When is it appropriate to use proteome size as a measure of complexity? What is your definition of complexity? When is the number of proteins in the proteome NOT a measure of increased information content or processing?
We thank the reviewer for this discussion. We realize that our use of the term complexity has been too liberal. To become more factual, we have removed most mentions to the concept of complexity from the manuscript. It is much more direct to speak about analyzing the correlation of the use of disorder to organismic properties: proteome size and number of cell types. We have added context on why we tried to find a correlation between fraction of disorder and proteome size though in section 3.2: “Since disorder in protein sequences has been associated to the emergence of cellular complexity [11] we wondered if there could be a correlation between fraction of disorder and proteome size, which is a possible measure of organismic complexity, following [12].”
The correlation of disorder with the number of cell types also seems superficial and overlooks the main point of the data.
We include now the significance of the correlations reported for Euteleostomi with the number of cell types: IDP fraction correlation r = 0.56, p-value = 0.04, IDR fraction correlation r = 0.71, p-value = 0.006. We believe that these linear fits merit being reported. Unfortunately, the figure was stretched horizontally, which flattened the slopes indicated in the figure, impacting negatively the observation. We now display the panels side by side instead.
IDP fraction and IDR fraction have the same dependence (slope) on the number of cell types. Is this an accident of the way the two fractions are calculated or does it have some biological meaning?
Since we found significant correlation between IDP and IDR fraction (Figure 2a), the graphs shown in Figure 3a and 3b are very similar, as expected.
On page 6 line 148 the authors state: "The results from our general study above indicate that while there might be a relation between disorder and complexity, there are likely other functions in disordered regions related to environment or ecology."
I don't know about this. Seems like a leap. There is some relation but the variability in cellular environment can reinforce, cancel, or offset variability in ecology.
We have toned down this statement and corrected the use of the term complexity making it more specific by adding “organismic” complexity: “Our general study above discussed a possible relation between disorder and organismic complexity. However, examples such as Halobacteria suggest that there are likely other functions in disordered regions that could be related to environmental factors.”
On page 6 lines 156-173 should be moved to the methods.
We have moved the technical details from this paragraph to the methods section as appropriate.
The meaning of page 6 lines 186-188 is not clear. What does this mean? Which species have IDP fractions close to 1? This is not apparent from the data in Figure 1. According to the definition of IDP fraction you used in Figure 1 a value of 1 would mean that every protein in the proteome has a >/= IDR of 30 residues.
We are very sorry because in the corresponding sentence we erroneously indicated “cluster #5” instead of correctly pointing to “cluster #3”. We have corrected this mistake and hope that with the clarification above on the meaning of IDP fraction and IDR fraction this sentence is now clear.
Reviewer 2 Report
Dear Editor,
The manuscript entitled “Evolutionary study disorder in protein sequences” describes the distribution of intrinsically disordered proteins (IDPs) or regions (IDRs) in proteomes belonging to different species. In addition, Authors compared orthologs form five different organisms (human, Mus musculus, Danio rerio, Drosophila melanogaster and Saccharomyces cerevisiae) to investigate the association of protein function to the evolutionary emergence of disorder in relation to cellular complexity.
The paper needs some changes to move beyond an observational description of the data generated.
My specific comments are below:
Abstract
Pag.1 Line 19. To aid the reading of this manuscript by non-biologist readers (e.g. physicist, chemist, etc.); Euteleostomi should be Euteleostomi (i.e. bony vertebrates).
Introduction
The role of disorder in evolution was previously reported in several works, please consider these references:
https://doi.org/10.1016/j.sbi.2011.02.005,
https://doi.org/10.3390/ijms19113315,
https://doi.org/10.1016/j.sbi.2011.03.014,
https://doi.org/10.1021/pr060171o
https://doi.org/10.1007/s00239-019-09921-4
https://doi.org/10.1007/s00018-017-2559-0
Pag.1 Line 32. IDP and IDRs present a peculiar aminoacidic composition, please describe them.
Pag.1. Line 34. IDPs and IDRs play a key role also in liquid–liquid phase separation Please consider these references:
https://doi.org/10.1111/febs.15254,
https://doi.org/10.1074/jbc.m117.800466
Material and methods
This section is poor, and some protocols are described in the result section. For instance, paragraph 3.4 contains the description of the protocol used to study the evolutionary emergency of disorder and protein function. This description should be moved from the results section to material and methods.
Moreover, the methods used to perform cluster analysis and GO enrichment should be described in Material and methods and not in the results section.
Pag. 2 Line 63. Please, state the abbreviation CBRs.
Results
Pag.2 Line 70. Please use the journal style for the reference.
Pag.2 Line 72. Authors selected 10,695 proteomes, of these most are of bacterial origin (8968). Please, try to give an explanation.
Pag 2. Line 73. See the comment Pag.1 Line 19.
Pag 3. Line 92. The number of bacterial proteomes used by the authors is not clear, here they are 9008, while in line 72 they are 8968. Please clarify.
Pag. 5 Line 123. Figure 2b shown that in viruses, Archaea and Bacteria all the species present fraction IDP < 0.4 and fraction IDR < 0.2. However, the authors considered eukaryotic organisms with an IDP fraction> 0.6 and an IDR fraction> 0.2. It is unclear why the authors focus on these organisms and not those with an IDP fraction> 0.4 and an IDR fraction> 0.2. Please clarify.
Pag. 5 Line 135. I suggest adding information on the domains and species the Authors have considered.
Figure 3. I suggest adding a supplementary table with the organisms and the abbreviation used in this figure.
Paragraph 3.4. This paragraph should be extensively revised, Authors should be focus on the results and not on the protocol description.
Pag. 6 Line 157. Taxonomic names should be in italic.
Figure 4. What the authors means with “binding” (First line Cluster 2)?
Pag. 6 Line 185. Drosophila should be in italic.
Pag. 8 Line 214. Here there is a list of proteins belonging to each cluster (e.g. RPABC2, POLR1A, POLR3D and ERCC3 for cluster 2) which makes the paragraph difficult to read. I suggest describing only the main functions of these proteins and using Table S3 as reference.
Pag. 8 Line 221. Drosophila should be in italic.
Pag.9 Line 272. Please use the journal style for the reference.
Pag. 10 Line 281. Charged residues affect not only IDP compactness, but also their propensity to aggregate and perform phase separation. Please cite:
https://doi.org/10.1073/pnas.1516277113
https://doi.org/10.1073/pnas.0911107107,
https://doi.org/10.1016/j.bbagen.2017.09.002,
https://doi.org/10.1073/pnas.1304749110,
https://doi.org/10.3390/ijms21176208,
https://doi.org/10.3390/cells9010145
https://doi.org/10.1073/pnas.1706083114
Pag. 11 Line 304. Taxonomic names should be in italic.
Pag. 11 Line 317. Taxonomic names should be in italic.
Figure 7 and 8. Please add in the caption a brief description of binary patterns for representation.
Discussion
Discussion could be improved by considering previous works describing the role of disorder in protein evolution. See these references:
https://doi.org/10.1016/j.sbi.2011.02.005,
https://doi.org/10.3390/ijms19113315,
https://doi.org/10.1016/j.sbi.2011.03.014,
https://doi.org/10.1021/pr060171o
https://doi.org/10.1007/s00239-019-09921-4
https://doi.org/10.1007/s00018-017-2559-0
One of the conclusions of the work is that E and K-rich regions are evolutionary stable. However, in the discussion the Authors does not mention the importance of charge residues on IDPs structure and function please consider these references:
https://doi.org/10.1073/pnas.1516277113
https://doi.org/10.1073/pnas.0911107107,
https://doi.org/10.1016/j.bbagen.2017.09.002,
https://doi.org/10.1073/pnas.1304749110,
https://doi.org/10.3390/ijms21176208,
https://doi.org/10.1073/pnas.1706083114
Author Response
We thank the referee for the insightful comments, which have certainly helped to improve the manuscript.
The manuscript entitled “Evolutionary study disorder in protein sequences” describes the distribution of intrinsically disordered proteins (IDPs) or regions (IDRs) in proteomes belonging to different species. In addition, Authors compared orthologs form five different organisms (human, Mus musculus, Danio rerio, Drosophila melanogaster and Saccharomyces cerevisiae) to investigate the association of protein function to the evolutionary emergence of disorder in relation to cellular complexity.
The paper needs some changes to move beyond an observational description of the data generated.
My specific comments are below:
Abstract
Pag.1 Line 19. To aid the reading of this manuscript by non-biologist readers (e.g. physicist, chemist, etc.); Euteleostomi should be Euteleostomi (i.e. bony vertebrates).
Done
Introduction
The role of disorder in evolution was previously reported in several works, please consider these references:
https://doi.org/10.1016/j.sbi.2011.02.005,
https://doi.org/10.3390/ijms19113315,
https://doi.org/10.1016/j.sbi.2011.03.014,
https://doi.org/10.1021/pr060171o
https://doi.org/10.1007/s00239-019-09921-4
https://doi.org/10.1007/s00018-017-2559-0
Done
Pag.1 Line 32. IDP and IDRs present a peculiar aminoacidic composition, please describe them.
Done
Pag.1. Line 34. IDPs and IDRs play a key role also in liquid–liquid phase separation Please consider these references:
https://doi.org/10.1111/febs.15254,
https://doi.org/10.1074/jbc.m117.800466
Done
Material and methods
This section is poor, and some protocols are described in the result section. For instance, paragraph 3.4 contains the description of the protocol used to study the evolutionary emergency of disorder and protein function. This description should be moved from the results section to material and methods.
Moreover, the methods used to perform cluster analysis and GO enrichment should be described in Material and methods and not in the results section.
We have moved the technical material from this section to the Material and Methods section as requested.
Pag. 2 Line 63. Please, state the abbreviation CBRs.
Done
Results
Pag.2 Line 70. Please use the journal style for the reference.
Thanks for noticing the formatting error. This text has been removed following a query of reviewer #1.
Pag.2 Line 72. Authors selected 10,695 proteomes, of these most are of bacterial origin (8968). Please, try to give an explanation.
We added this sentence: “Of note, the over-representation of bacterial proteomes is due to the high number of bacterial sequencing projects, which are facilitated by the accessibility and small genome size of bacteria [30].”
Pag 2. Line 73. See the comment Pag.1 Line 19.
Done
Pag 3. Line 92. The number of bacterial proteomes used by the authors is not clear, here they are 9008, while in line 72 they are 8968. Please clarify.
Thank you for spotting this inconsistency, which we have corrected. The correct value is 8968.
Pag. 5 Line 123. Figure 2b shown that in viruses, Archaea and Bacteria all the species present fraction IDP < 0.4 and fraction IDR < 0.2. However, the authors considered eukaryotic organisms with an IDP fraction> 0.6 and an IDR fraction> 0.2. It is unclear why the authors focus on these organisms and not those with an IDP fraction> 0.4 and an IDR fraction> 0.2. Please clarify.
We chose those cut-offs because those species stand out from the central cloud of data points. We have added this missing explanation to the manuscript.
Pag. 5 Line 135. I suggest adding information on the domains and species the Authors have considered.
The set of 44 species and annotations were taken from reference Schad et al. Genome Biology 2011. We have extended the corresponding sentence to make it more explicit.
Figure 3. I suggest adding a supplementary table with the organisms and the abbreviation used in this figure.
We thank the referee for noticing that this information was missing. We have done as requested.
Paragraph 3.4. This paragraph should be extensively revised, Authors should be focus on the results and not on the protocol description.
We have moved the technical material from this section to the Material and Methods section as appropriate.
Pag. 6 Line 157. Taxonomic names should be in italic.
Done
Figure 4. What the authors means with “binding” (First line Cluster 2)?
That is a Gene Ontology term as indicated in the caption. The current definition will be always available at multiple web sites that distribute the Gene Ontology. Now it is: The selective, non-covalent, often stoichiometric, interaction of a molecule with one or more specific sites on another molecule.
Pag. 6 Line 185. Drosophila should be in italic.
Done
Pag. 8 Line 214. Here there is a list of proteins belonging to each cluster (e.g. RPABC2, POLR1A, POLR3D and ERCC3 for cluster 2) which makes the paragraph difficult to read. I suggest describing only the main functions of these proteins and using Table S3 as reference.
We would like to keep the protein names because finding this information in the table is not too easy and also because this could increase the impact of the manuscript by attracting the attention of researchers interested in those proteins.
Pag. 8 Line 221. Drosophila should be in italic.
Done
Pag.9 Line 272. Please use the journal style for the reference.
Done
Pag. 10 Line 281. Charged residues affect not only IDP compactness, but also their propensity to aggregate and perform phase separation. Please cite:
https://doi.org/10.1073/pnas.1516277113
https://doi.org/10.1073/pnas.0911107107,
https://doi.org/10.1016/j.bbagen.2017.09.002,
https://doi.org/10.1073/pnas.1304749110,
https://doi.org/10.3390/ijms21176208,
https://doi.org/10.3390/cells9010145
https://doi.org/10.1073/pnas.1706083114
Done.
Pag. 11 Line 304. Taxonomic names should be in italic.
Done
Pag. 11 Line 317. Taxonomic names should be in italic.
Done
Figure 7 and 8. Please add in the caption a brief description of binary patterns for representation.
Done
Discussion
Discussion could be improved by considering previous works describing the role of disorder in protein evolution. See these references:
https://doi.org/10.1016/j.sbi.2011.02.005,
https://doi.org/10.3390/ijms19113315,
https://doi.org/10.1016/j.sbi.2011.03.014,
https://doi.org/10.1021/pr060171o
https://doi.org/10.1007/s00239-019-09921-4
https://doi.org/10.1007/s00018-017-2559-0
Done
One of the conclusions of the work is that E and K-rich regions are evolutionary stable. However, in the discussion the Authors does not mention the importance of charge residues on IDPs structure and function please consider these references:
https://doi.org/10.1073/pnas.1516277113
https://doi.org/10.1073/pnas.0911107107,
https://doi.org/10.1016/j.bbagen.2017.09.002,
https://doi.org/10.1073/pnas.1304749110,
https://doi.org/10.3390/ijms21176208,
https://doi.org/10.1073/pnas.1706083114
Done
Round 2
Reviewer 1 Report
No additional changes are required.